

# The origin of *Oxalis corniculata* L.

Quentin J. Groom[1], Jan Van der Straeten[2] and Ivan Hoste[1]

[1] Meise Botanic Garden, Meise, Belgium
[2] Laboratory of Plant Science and Nature Management, Vrije Universiteit Brussel, Brussels, Belgium

## ABSTRACT

**Background:** *Oxalis corniculata* L. is a weed with a world-wide distribution and unknown origin. Though it belongs to a section of the genus from South America, the evidence that this species came from there is weak.

**Methods:** We reviewed the evidence for the origin of *O. corniculata* using herbarium specimens, historic literature and archaeobotanical research. We also summarized ethnobotanical literature to understand where this species is most used by humans as a medicine.

**Results:** Despite numerous claims that it is native to Europe there is no strong evidence that *O. corniculata* occurred in Europe before the 15th century. Nor is there reliable evidence that it occurred in North or South America before the 19th century. However, there is direct archaeobotanical evidence of it occurring in south–east Asia at least 5,000 years ago. There is also evidence from historic literature and archaeobotany that it reached Polynesia before European expeditions explored these islands. Examination of the traditional use of *O. corniculata* demonstrates that is most widely used as a medicine in south–east Asia, which, while circumstantial, also points to a long association with human culture in this area.

**Discussion:** The most likely origin for *O. corniculata* is south–east Asia. This is consistent with a largely circum-Pacific distribution of section *Corniculatae* of *Oxalis*. Nevertheless, it is likely that *O. corniculata* spread to Europe and perhaps Polynesia before the advent of the modern era through trade routes at that time.

Corresponding author
Quentin J. Groom,
quentin.groom@plantentuinmeise.be

## INTRODUCTION

It is often stated that the origin of *Oxalis corniculata* L. is obscure or unknown. The species was first described by Carl Linnaeus from material from the Mediterranean region and some authors suggest that this region is the native range (*Ridley, 1930*; *Young, 1968*; *Lourteig, 1979*; *Gray, 2011*). Currently, *O. corniculata* has the third largest distribution of any vascular plant species (*Pyšek et al., 2017*), having been introduced to practically every country. It has even been introduced to sub-Antarctic islands (*Frenot et al., 2005*). This global distribution and its relatively early spread has contributed to the difficulty of identifying its native range. Furthermore, it is almost constantly associated with anthropogenic disturbed habitats which makes its native range and the native status of local populations more difficult to determine.

The designation of a native range can be important from a policy perspective, particularly where this relates to the control of invasive species and the avoidance of further spread. It is also required from a scientific perspective, to understand the evolution of section *Corniculatae* DC. of the genus *Oxalis*. *Vaio et al. (2013)* fixed the origin of section *Corniculatae* to South America using a molecular phylogenetic approach.

How this section subsequently colonized the world is an interesting question of evolution and dispersal. Molecular phylogenetics can only indicate where the ancestors of a taxon originate, but physical evidence is needed to establish where each taxon in that phylogeny is native. For most species the current distribution provides such physical evidence, but this is unreliable in a widely dispersed species such as *O. corniculata*. In some cases, by combining genetic and historical physical evidence, the distributions of such species can been explained. For example, the New World distributions of the domestic dog (*Canis lupus familiaris* L., 1758) and the bottle gourd (*Lagenaria siceraria* (Molina) Standl.) have been explained this way (*Savolainen et al., 2002*; *Erickson et al., 2005*; *Larson et al., 2012*).

*Oxalis corniculata* is highly persistent in the horticulture industry and is frequently found in gardens and as a hitchhiker in plant pots in nurseries and garden centres (*Neal & Derr, 2005*). Its explosive capsules, sticky seeds, extended flowering period and short generation time make it a successful colonizer and persistent weed. Although it is not a strong competitor, its enormous range and abundance must multiply its overall impact as a weed. Furthermore, although *O. corniculata* is the weediest and most widespread of the species in the section *Corniculatae*, several other species in the section have spread beyond their native ranges. These include *O. stricta* L. and *O. dillenii* Jacq. from North America and *O. exilis* A.Cunn. from Australasia (*Young, 1968*).

The taxonomy of *O. corniculata* is complex. It is variable cytologically and genetically, but is also phenotypically plastic (*Mathew, 1958*; *Vaio et al., 2013*). Its taxonomy is complicated by the description of many subspecific taxa and other species now considered to be synonyms (*Lourteig, 1979*). Its species limits vary depending on the authority, notably *Eiten (1963)* who tended to lump taxa and *Lourteig (1979)* who tended to split them. Furthermore, there has been taxonomic confusion of *O. corniculata* with closely related species, such as *O. stricta* and *O. dillenii*. In this paper we follow the nomenclature of *Watson (1989)* and *Eiten (1963)*. One of the notable infraspecific taxa of *O. corniculata* is the purple-leaved variety *atropurpurea* of horticultural origin (*Planchon, 1857*).

Although much has been written about *O. corniculata*, the corpus has never been put together to form conclusions as to its origin. Here, we review herbarium specimens, historic literature and archaeobotanical research to reach a conclusion on the biogeography of *O. corniculata*. Using herbarium specimens and literature we can document the presence and introduction of the species during the Modern Era. Archaeobotanical evidence helps us push the boundaries of our knowledge further back, but we also draw on other lines of evidence, some direct, some circumstantial.

## METHODS

First records of *O. corniculata* were gathered from literature on biodiversity and herbarium specimens. National and regional Floras were consulted, but also the literature indexed by the Biodiversity Heritage Library and the digitised manuscripts of the British Library. The monograph of *Lourteig (1979)* was also critical as she toured many of the major herbaria identifying specimens. For mapping earliest records we followed the specimens she identified as *O. corniculata* and its varieties, but excluded species such as *O. radicosa* A.Rich. and *O. procumbens* Steud. ex A.Rich. that may be included within *O. corniculata* by other authors. Many specimens and observations were also found by searching the Global Biodiversity Information Facility. However, due to the presence of related species, such as *O. dillenii* in North America and other species elsewhere, we treated observations in literature references cautiously if not combined with evidence.

A literature review on the ethnopharmacology of *O. corniculata* was conducted initially through Google Scholar, the Web of Science and the online catalogue of the library of Meise Botanic Garden, with searches on the keywords "oxalis," "corniculata," "medicinal," "ethnobotany" and "pharmacology." A particular effort was made to search ancient herbals for mentions of any plant that might be *O. corniculata*. Old Latin names for *O. corniculata* include *Oxys luteo flore*, *Lotus urbana* and *Trifolium acetosum corniculatum*, but there are also old vernacular names for *Oxalis* sp. such as *alleluja*, *pain de coqu* and *sawerklee*.

Each reference was documented, together with the ailment it was used to cure and the locality. *O. corniculata* was reportedly used in the treatment of many ailments including diarrhoea, dysentery, fever, scurvy, stomach aches, wounds, snakebite and scorpion stings. Once the initial literature search was exhausted we used targeted searches including these ailments as keywords to find more references. Finally, every paper from 1979 onward in the Journal of Ethnopharmacology was reviewed for mentions of *O. corniculata* (*Van Der Straeten & Groom, 2018*).

Localities for first records and ethnopharmacological uses were recorded using the level 4 codes in the World Geographical Scheme for Recording Plant Distributions (*Brummitt, 2001*). However, if the description of the locality was too indistinct the locality was recorded using the ISO 3166:1988 country codes.

## RESULTS

### A continental summary of the distribution of *Oxalis corniculata*
#### Europe and North Africa

Europe has five extant species of *Oxalis* section *Corniculatae*, *O. corniculata*, *O. exilis*, *O. stricta*, *O. dillenii* and *O. filiformis* Kunth (syn. *O. ferae* L.Llorens, Gil & C.Cardona). Of these, *O. exilis* and *O. filiformis* are 20th century introductions with relatively narrow distributions (*Llorens et al., 2005*; *Young, 1968*). Owing to the presence of closely related species in Europe it is necessary to consider confusion of their identification in observations. Unless a complete specimen is available it is difficult to distinguish these taxa from one another, particularly if only fragments, such as seeds, are available. Taxonomic and nomenclatural confusion is a constant theme in the literature of this

section (*Eiten, 1955*; *Watson, 1989*). Therefore, before investigating the situation of *O. corniculata* it is important to establish the time of introduction of *O. stricta* and *O. dillenii* to Europe.

It was not until 1680 that Morison described a species with an upright habit that we now refer to as *O. stricta* (*Morison, 1680*). However, this upright species was described from seed sent to him from Virginia and was not from Europe. By the mid-19th century the upright species (*O. stricta*) was so well established in Europe that *Jordan (1854)* considered that it must be a native species and described *O. europaea* Jord. He did so based purely on the geographic grounds that the Linnaean *O. stricta* was North American. He did not give any morphological characters that separate them. Herbarium specimens do exist of *O. stricta* collected in Europe before 1854, but these are few and only date from earlier in the 19th century (Table 1) (*Lourteig, 1979*).

*O. dillenii* was described and illustrated by *Dillenius (1732)* and was formally named later by *Jacquin (1794)* with direct reference to Dillenius. However, this was from material of North American origin. It was not until later that *O. dillenii* was recognised as naturalized in Europe. Perhaps the first record is that of *Jordan (1854)* who named *O. navieri* Jord., which is now considered a synonym for *O. dillenii*. Therefore, we agree with all modern authors that both *O. stricta* and *O. dillenii* are 19th century introductions and we can assume that earlier European observations of caulescent yellow-flowered *Oxalis* are of *O. corniculata*.

The earliest clear mention of a yellow-flowered caulescent *Oxalis* in European literature is from northern Italy in the 16th century (*Turner, 1568*). The earliest herbarium specimens are also from northern Italy, from Florence in 1600 and Bologna between 1551 and 1600 (Table 1). *Watson (1989)* reviewed the taxonomic situation of *O. corniculata* in Europe and cites Renaissance botanists of the 16th and 17th century from Belgium, France, Germany, the Netherlands, Spain and Switzerland. Therefore, there is good evidence, including accurate drawings, that *O. corniculata* occurred in Europe at least from the 16th century.

Earlier than this the information in literature is ambiguous. We have found six illustrations of *Oxalis* (*Alleluia* or *Panis Cuculi* in Latin) in medieval herbals from the 13th to 15th centuries. In each case only one taxa is illustrated. In Europe we would expect at least one taxon, the native *O. acetosella* L. This species is quite distinct from *O. corniculata* in having white flowers, short globular pods, a scaly creeping rhizome and no stem. The illustration in the 15th century Codex Bellunensis and an unnamed codex have no flowers, but depict the scaly rhizome and the lack of stem, which suggests this is *O. acetosella* (*British Library, 2019a*; *Schoenberg Center, 2019*). The illustration in Giovanni Cadamosto's manuscript of the late 15th or early 16th century ambiguously shows a part caulescent and part acaulescent plant, (*British Library, 2019b*). The flowers of this illustration are in bud and are of little diagnostic use. A similarly ambiguous picture is in a codex dated 1475–1525, also from Italy (*University of Vermont Libraries, 2019*). Another herbal from circa 1440 shows a caulescent *Oxalis* with small yellow flowers in umbels. This illustration is close to the characters of *O. corniculata* and originates from Lombardy in northern Italy (*British Library, 2019c*). Finally, the oldest illustration is

**Table 1 Notable early specimens of *Oxalis corniculata* and *O. stricta*.**

| Taxon | Collector name | Collector number | Date | Country | Catalogue number & HTTP URI | Notes |
|---|---|---|---|---|---|---|
| *Oxalis stricta* | C.E.Broome | *s.n.* | 1836 | Belgium | http://data.rbge.org.uk/herb/E00874849 | An example of a European specimen collected before *O. europaea*, was described. |
| *Oxalis stricta* | *s.c.* | *s.n.* | 1835 | Germany | http://data.rbge.org.uk/herb/E00874850 | An example of a European specimen collected before *O. europaea*, was described. |
| *Oxalis corniculata* | Joachim Burser | *s.n.* | 1600 | Italy | UPS:BOT:V-175221 http://www.gbif.org/occurrence/328193149/verbatim | An early specimen from Europe. |
| *Oxalis corniculata* | Ulisse Aldrovandi | *s.n.* | 1551–1600 | Italy | Volume 8, page 134 | An early specimen from Europe. |
| *Oxalis corniculata* | Carl Peter Thunberg | 11118 | 1772–1775 | South Africa | UPS:BOT:V-086842 | The earliest specimen from sub-Saharan Africa (Lectotype of *O. repens* Thunb.) |
| *Oxalis corniculata* | Christian Friedrich Ecklon & Carl L.P. Zeyher | *s.n.* | 1830 | South Africa | PRE0453493-0 https://www.gbif.org/occurrence/462122048 | An early specimen from South Africa. |
| *Oxalis corniculata* (as *O. procumbens*) | Wilhelm Schimper | 1165 | 1838-02-05 | Ethiopia | TUB-001752 http://id.snsb.info/snsb/collection/20919/29698/20739 | An early specimen from East Africa. |
| *Oxalis corniculata* | Wilhelm Schimper | *s.n.* | 1837-06 | Ethiopia | TUB-001748 http://id.snsb.info/snsb/collection/112371/171149/113437 | An early specimen from East Africa. |
| *Oxalis corniculata* | Carlo L.G. Bertero | 494 | 1827–1830 | Chili | MO-1063304 | An early specimen from South America. |
| *Oxalis corniculata* | Prince Maximilian of Wied-Neuwied | *s.n.* | 1815–12 | Brazil | M-0153326 | An early specimen from South America, São Bento Monastery, Brazil. |
| *Oxalis corniculata* | George Gardner | 345 | 1837–08 | Brazil | MO-1063334 | An early specimen from South America. Cultivated. |
| *Oxalis corniculata* | Pehr Osbeck | *s.n.* | 1751 | China | SBT:H:2.4.9.67 | An early specimen from China. |

Notes:
  Cited and early specimens of *Oxalis* section *Corniculatae* from Europe, Africa, South America and China. N.B. *O. europaea*, which was described from European specimens, is considered a synonym of *O. stricta*, which was described from North American material.

in the *Tractatus de herbis* (circa 1280–1350) (*British Library, 2019d*). This is a rather confusing illustration. It shows a horizontal creeping plant, rooting at the nodes, which could be *O. acetosella* or *O. corniculata*. The presumed flowers are small and erect, like *O. corniculata*, but solitary like those of *O. acetosella*. There are other four-pointed structures, which are presumably fruits. However, they do not resemble *Oxalis* fruits.

A considerable number of archaeobotanical studies have been conducted in Europe, but remains of *Oxalis* section *Corniculatae* remain rare. The RADAR database in

the Netherlands contains four records, the earliest of which are from a site that spans the 17th and 18th centuries (*Van Haaster & Brinkkemper, 1995*). In Ferrara, Italy, remains of a single seed were found in a waste pit dating from the second half of the 15th century, which is notable because of the pre-Columbian dating (*Bosi, 2000*, *Bosi et al., 2009*). Although a single seed, its location is close to the specimen, illustration and literature reports from the 15th and 16th centuries, giving credence to this isolated observation. We have found no earlier archaeological evidence, even though some extensive studies have been conducted. *Rinaldi, Mazzanti & Bosi (2013)* summarize the seeds found in excavations in Modena, in northern Italy from the 2nd century BC to the 6th century AD. They list more than 400 taxa from 200,000 remains, including many weeds, but did not find *O. corniculata*. Likewise a similar study in Lleida, Spain, covering the period between the 2nd century BC and the 11th century AD report no *O. corniculata*, despite recovering almost 600,000 seeds and fruits.

We have found two other archaeological reports of *O. corniculata* in Europe, one from Neolithic Germany in the 6th century BC and the other from Visigothic Spain in the 6th-7th century AD (*Herbig et al., 2013*; *Ollich et al., 2014*). The reports are only of two and one seed, respectively, and there are no illustrations. Given the small number of seeds, the lack of supporting evidence and the potential for later contamination or misidentification, we have discounted these reports.

In all, we found no convincing evidence for the presence of *O. corniculata* in Europe earlier than the 15th century. Neither is there much information from North Africa, which has been in constant communication with southern Europe for many centuries. Preserved *Oxalis* seeds have been reported from late neolithic Hyrax middens in the Hoggar Mountains in the Sahara (*Barakat, 1995*). However, no illustration is provided and there is no way to verify this unlikely claim. A more reliable source is from excavations at ancient Carthage in modern day Tunisia (*Van Zeist, Bottema & Van Der Veen, 2001*). Here, preserved *Oxalis* seeds were reported from sediment from approximately the 4th century BC to the mid-6th century AD. However, there is again no illustration and we have been unable to access vouchers. The earliest direct observation from North Africa is from 1751 (*Hasselquist, 1766*). This was near Damietta, Egypt, in the Nile delta.

In conclusion, there is good evidence of *O. corniculata* being in Europe from the 16th century onward. Despite considerable taxonomic confusion, there is no evidence of *O. stricta* and *O. dillenii* occurring here much before the beginning of the 19th century, presumably as an introduction from North America, as is generally assumed. Earlier than this, despite a copious corpus of botanical, medicinal and archaeobotanical literature, the evidence is weak, leaving an open question about the long-term existence of *O. corniculata* in Europe. The observation of *O. corniculata* in Carthage may indicate that *O. corniculata* was present in the Mediterranean basin during the Iron Age, though more evidence than this would be preferable. Therefore there is little support for *O. corniculata* being native to Europe.

### Sub-Saharan Africa

Today *O. corniculata* is a common weed in sub-Saharan Africa (*Holm et al., 1991*). However, *Hooker & Bentham (1849)* specifically noted its absence from the western

intertropical coast, where it is present today (*Hutchinson & Dalziel, 1954*; *Holm et al., 1991*). The earliest sub-Saharan records are of Thunberg from near Cape Town, South Africa, circa 1772 (Table 1). Thunberg described its habitat as "*Crescit prope urbem locis aquosis et in hortis*" (Growing in wet ground near the city gardens) (*Thunberg, 1823*). Thunberg described *O. repens* Thunb. from these specimens, which has subsequently been synonymized with *O. corniculata* (*Lourteig, 1979*). The next specimens are from the 1830s from South Africa and Ethiopia (Table 1), but there are few records from sub-Saharan Africa until the 20th century and many records are of *O. corniculata* as an agricultural weed.

In Ethiopia *Richard, Petit & Quartin-Dillon (1847)* claimed three species, *O. corniculata* and two new species *O. procumbens* and *O. radicosa*. *Lourteig (1979)* accepted *O. procumbens* and *O. radicosa* as distinct species. *Oxalis procumbens* is said to grow in fields and fallow land at altitudes above 1,000 m in tropical Africa and *O. radicosa* has a larger distribution spanning much of Africa, Asia and Australasia. Examples of specimens are in Table 1. Many other authors, at least in Africa, have treated these species as synonyms of *O. corniculata* (*Engler, 1894*; *Madagascar Catalogue, 2017*). If one accepts that *O. procumbens* and *O. radicosa* are distinct species in Africa, then there is little reason to think that *O. corniculata* s.s. is native to sub-Saharan Africa. However, if these species are considered to be part of *O. corniculata* s.s. then it is perhaps native to the east coast of Africa and Madagascar and also to high altitude areas of tropical Africa. However, the lack of systematic collecting in Africa before the 20th century means that the indigenousness of specimens attributed to *O. procumbens* and *O. radicosa* can also not be confirmed.

The lack of data from much of Africa make it difficult to draw conclusions and this is further complicated by the ambiguity of the species concept of *O. corniculata*. The history of *O. corniculata* in sub-Saharan Africa will probably remain obscure without further evidence or discoveries.

### Australasia

Australasia has several native species in section *Corniculatae*, but *O. corniculata* s.s. is considered introduced (*Webb, Sykes & Garnock-Jones, 1988*; *Conn, Jeanes & Richards, 1999*; *Gray, 2011*). Native taxa are *O. novae-caledoniae* R.Knuth & Schltr., *O. exilis*, *O. rubens* Haw, *O. perennans* Haw., *O. chnoodes* Lourteig, *O. radicosa* and *O. thompsoniae* B.J.Conn & P.G.Richards. Unlike authors of African Floras, authors of Australasian Floras accept *O. radicosa* as a distinct species. Australasian species live in a wide range of natural and anthropogenic habitats and although all are close in morphology to *O. corniculata*, all have distinct morphological features that separate them. Nevertheless, confusion of specimens and observations of *O. corniculata* with the native taxa must always be considered. Study of the Australasia situation is complicated because Eiten considered all Australasian members of the section *Corniculatae* to be part of *O. corniculata* s.l. and many specimens in herbaria retain his determinations (*Eiten, 1963*).

Of possible native species, the closest species to *O. corniculata* in Australasia is *O. thompsoniae* that has recently been described (*Conn & Richards, 1994*). It has a weedy habitat and is only known from collections made from the late 20th century.

*Conn & Richards (1994)* suggest it may have been introduced to Australasia, but is of unknown origin. Likewise it may have evolved recently in Australasia from *O. corniculata*. Nevertheless, all reliable determinations of *O. corniculata* s.s. in Australasia date from the mid to late 20th century.

In summary, the prevailing consensus that *O. corniculata* is a recent introduction to Australasia seems to be consistent with the available evidence. However, the similarities of Australasian species with *O. corniculata* perhaps indicate a close evolutionary relationship.

### North America

The Flora of North America lists several native members of *Oxalis* section *Corniculatae*. These are *O. illinoensis* Schwegman, *O. macrantha* (Trel.) Small, *O. grandis* Small, *O. colorea* (Small) Fedde, *O. stricta*, *O. suksdorfii* Trel., *O. dillenii*, *O. albicans* Kunth, *O. pilosa* Nutt. ex Torr. & A.Gray, *O. californica* (Abrams) R.Knuth and *O. texana* (Small) Fedde. Furthermore, in Mexico, Central America and the Caribbean there are other species, *O. filiformis*, *O. rugeliana* Urb. and *O. thelyoxys* Focke (*Nesom, 2017*).

*Oxalis corniculata* s.s. is considered an introduction to North America (*Nesom, 2009*). In Central America and the Caribbean the earliest herbarium specimens date from the 1820s (*Lourteig, 1979*). However, an illustration in *Hernández (1651)* has been interpreted as being *O. corniculata* (*Pico & Nuez, 2000*), although the illustrated specimen cannot be confidently distinguished from other North American taxa, such as *O. albicans*.

We found no convincing evidence that conflicts with the widely held view that *O. corniculata* has been introduced to North America.

### South America

Several species in section *Corniculatae* have been reported in South America such as *O. bisfracta* Turcz., *O. calachaccensis* R.Knuth, *O. corniculata*, *O. filiformis* and *O. sexenata* Savigny (*Eiten, 1963*). Also, others, such as *O. dumetorum* Barnéoud and *O. conorrhiza* Jacq. can be included under a broader circumscription of section *Corniculatae* (*Vaio et al., 2013*). *Vaio et al. (2013)* used a molecular phylogeny to trace the evolutionary roots of section *Corniculatae* to South America. However, establishing whether *O. corniculata* s.s. is native to South America is complicated because of the other closely related species present on the continent. Where observations are not supported by specimens it is impossible to know if they have been correctly identified. The earliest specimens we have found of *O. corniculata* from South America (Brazil) are from 1815 to the 1830s (Table 1). However, in both cases the plants were either cultivated or weeds of cultivation.

*Vaio et al. (2013)* showed that *O. corniculata* is genetically close to the Bolivian species *O. calachaccensis*. In fact, *O. calachaccensis* does not possess any character that uniquely distinguishes it from *O. corniculata* (*Eiten, 1963*). *O. calachaccensis* was described by *Knuth (1915)* using material he collected in 1911 from La Paz, Bolivia and one is left to assume it is native to that location. However, as *Vaio et al. (2013)* indicate, the taxonomic status of *O. calachaccensis* is doubtful.

No earlier literature or archaeobotanical evidence has been found for *O. corniculata* s.s in South America. Even though *O. corniculata* has South American ancestry, there is no evidence for the species being in South America before the 19th century.

### Asia

*Oxalis corniculata* is widespread in Asia and is an important weed in countries such as India and China (*Reddy, 2008*; *Liu & Watson, 2008*). The authors of Asian Floras, such as those from China and Japan, do not commit to whether *O. corniculata* is native or introduced, but they conclude that the widespread dispersal by humans has made its origin obscure (*Ōi, Meyer & Walker, 1965*; *Liu & Watson, 2008*).

Nevertheless, there are several archaeobotanical reports of *O. corniculata* seed from China. The earliest of these are from the Majiabang culture (Hunan, 7000–5800 BP; 16 seeds in two excavations) and Daxi culture (Jiangsu, 7,000–5,300 BP; 17 seeds) (*Nasu et al., 2012*; *Qiu et al., 2016*). Then are reports from Henan Province 4,950–4,450 BP where 74 seeds were found (*Deng et al., 2015*). There are also reports of multiple preserved seed from the Huangguashan period in Fujian province from more than 3,000 BP; from the Bronze Age in Yunnan province, and from during the Tang dynasty in Lantau (*Atha, 2012*; *Yao et al., 2015*; *Deng et al., 2017*). The paper of *Nasu et al. (2012)* has a particularly convincing illustration of a seed. At all of these sites there was evidence of early agriculture where *O. corniculata* may have been a weed.

Apart from *O. corniculata* the only other possibility is that these are seeds of *O. stricta*, which is believed to be native to China (*Liu & Watson, 2008*). However, this Flora of China describes *O. stricta* as growing in "Forests, ravines; 400–1,500 m" in Guangxi, Hebei, Henan, Hubei, Jiangxi, Jilin, Liaoning, Shanxi, Zhejiang. This is not consistent with the presence of this *Oxalis* in either Hunan, Fujian, Yunnan or Jiangsu province as an agricultural weed.

Also in Asia, *Oxalis* seeds have been found in Japan from the Early Yayoi period (ca. 2,820–2,530 BP); in the Philippines in the 1st millennium BC and in Vietnam between 3,450 and 3,250 BP (*Paz, 2005*; *Nasu & Momohara, 2016*; *Castillo et al., 2018*). *Castillo et al. (2018)* provide a photograph of the single seed they found. *Nasu & Momohara (2016)* indicate that they found between 11 and 58 seeds across two different excavations. *O. stricta* has neither been found in the Philippines nor Vietnam, so it seems reasonable to assume these are seeds of *O. corniculata*.

There is early documentary evidence of the presence of *O. corniculata* in China from 1751 (*Osbeck, Torén & Ekeberg, 1771*). This is from Guangdong, outside the known range of *O. stricta* in China and in a region where *O. corniculata* is common today. There is also a specimen from Osbeck in the Bergius Herbarium from 1751 and this has been identified by Lourteig as *O. corniculata* (Table 1). The distribution of first records illustrated in Fig. 1 demonstrates the cluster of early records from south–east Asia.

*Oxalis corniculata* is also the main food plant of the larvae of *Pseudozizeeria maha* Kollar, 1844 (pale grass blue butterfly) (*Saji et al., 2017*). This species is found across India, south–east Asia, China, North and South Korea and Japan. The presence of this largely monophagous insect suggests that the original native range of its host plant lies within Asia.

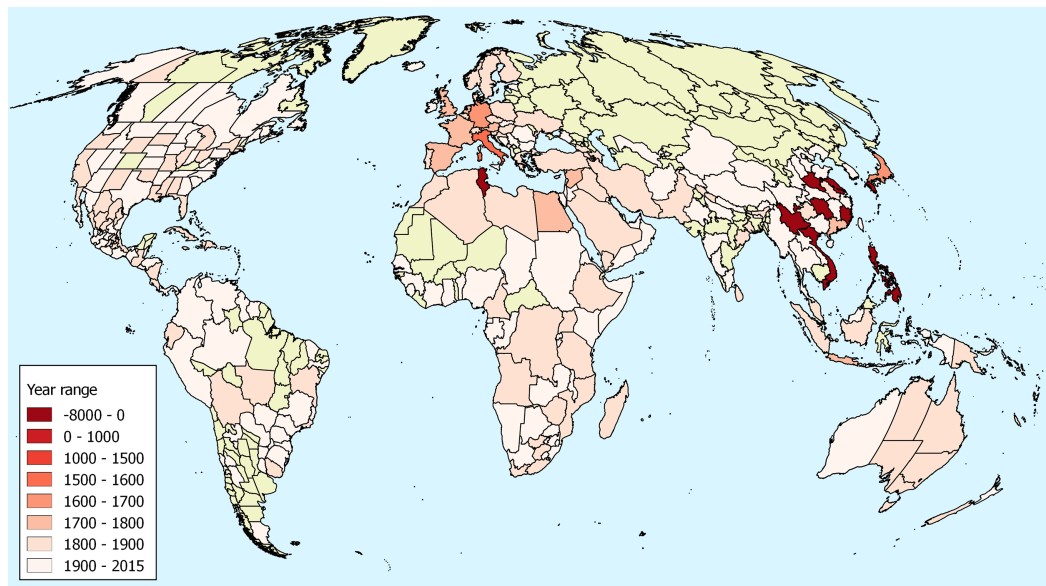

**Figure 1 First observations of *Oxalis corniculata*.** First observations of *Oxalis corniculata* globally based upon herbarium specimens, historic literature and paleobotanical research. The map uses the World Geographical Scheme for Recording Plant Distributions (*Brummitt, 2001*). Areas were no records were discovered are coloured in green. Note that the scale used is non-linear to allow visualization of the differences in modern introduction dates. The map uses a Mollweide projection. Details of these records have been deposited in an open repository (*Groom, 2018*).     

While it is an indirect indication, the use of plants in traditional medicine can perhaps indicate a long association of a plant with human culture. In our survey of the medicinal uses of *O. corniculata* we discovered a total of 182 publications containing 892 remedies for ailments at different locations. It is evident from this map that Asia is at the centre of where *O. corniculata* is used medicinally (Fig. 2).

In conclusion, there are several high-quality paleobotanical records of *O. corniculata* in eastern Asia and early documentary evidence. Furthermore, there is at least one insect that feeds on *O. corniculata* in the continent and there is ample documentation of the use of *O. corniculata* in traditional Asian medicine. Therefore, there are several lines of evidence for *O. corniculata* having a long presence in Asia, although the association of *O. corniculata* with human disturbance and agriculture has probably led to its further spread within Asia. This makes it difficult to point to a more specific area of origin. However, the archaeobotanical remains from eastern China, Japan and the Philippines do point to a long history in this area.

### Polynesia

*Oxalis corniculata* was present on Hawaii (1779) and Tahiti (1769) when they were explored during the scientific expeditions of Captain James Cook (*St John, 1978*). Indeed, archaeological excavations have found *O. corniculata* seed in sediment dated from between the 15th and 17th centuries (*McCoy, 1977*; *Allen, 1984*). Therefore, *O. corniculata* is either native to Polynesia or, as suggested by *Ridley (1930)*, was introduced by Polynesian colonizers.

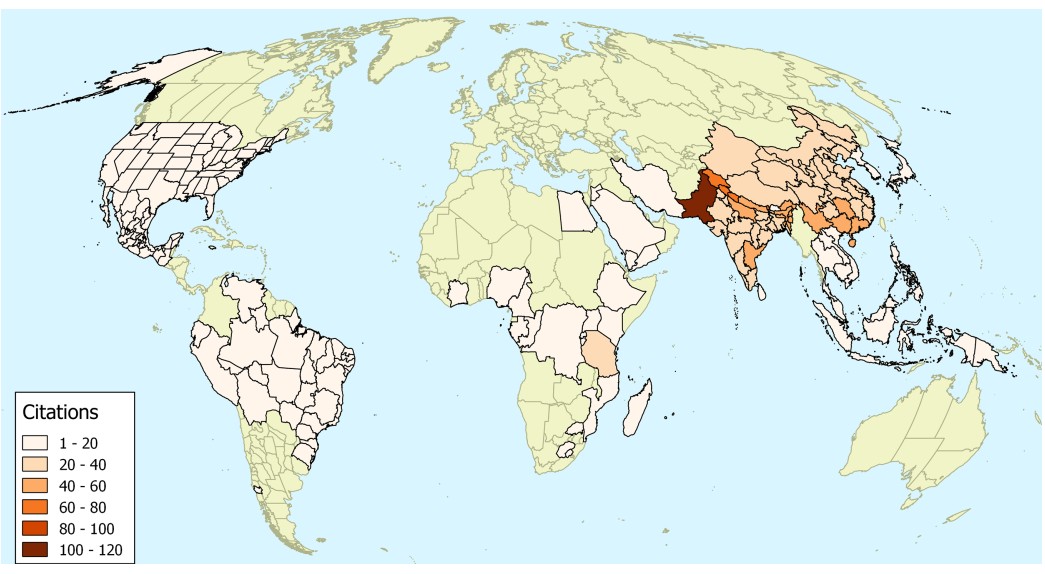

**Figure 2 The number of ethnopharmacological citations.** The number of ethnopharmacological citations for the uses of *Oxalis corniculata*. Areas with no reports of usage are coloured in green. The map uses the World Geographical Scheme for Recording Plant Distributions (*Brummitt, 2001*). The map uses a Mollweide projection.

## DISCUSSION

The concept of native range is fluid and difficult to define precisely. It is hard to separate human activities from the supposed "natural" distribution of the organism, particularly when the term is used with agricultural weeds. Rather than define nativeness precisely we have merely tried to seek evidence for the presence of *O. corniculata* during its global colonization.

*Vaio et al. (2013)* provides us with a fixed origin for the section *Corniculatae* in South America, but the further speciation in Australasia and North America indicates that dispersal of the section to those continents occurred hundreds of thousands of years before present. *Heibl & Renner (2012)* estimate that *O. corniculata* diverged from North American taxa (*O. stricta* and *O. dillenii*) about 10 million years ago. Therefore, there is no reason to presuppose that *O. corniculata* s.s. evolved in South America. Furthermore, additional resolution of the phylogeny of the section *Corniculatae* is needed. *Vaio et al. (2013)* did not include any of the known Australasian, Central American and Caribbean *Corniculatae* and only two of the North American species.

Although Europe has often been suggested as the origin of *O. corniculata*, there is no convincing evidence for its occurrence here earlier than the Renaissance. In contrast, there is clear evidence of *O. corniculata* in eastern Asia occurring for several thousand years. Archaeological evidence supports this conclusion. Furthermore, early specimens, literature and anthropogenic usage are consistent. This conclusion also makes sense regarding the proximity of its nearest relatives. These are native to Australasia, North America and South America. This means, if one accepts the eastern Asian origin of *O. corniculata*, that then *Oxalis* section *Corniculatae* has a largely circum-Pacific

distribution, which we find rather more parsimonious than one member of the section naturally occurring in relative isolation in Europe.

There is undoubtedly considerable bias in the publication of ethnobotanical literature and in the intensity of collection of specimens from different parts of the world. In general, native plants are more often used in traditional medicines, but that is probably because there are more available (*De Medeiros, Ladio & Albuquerque, 2013*). Plants are also specifically introduced for their medicinal properties. Nevertheless, these independent sources of evidence both focus on subtropical eastern Asia as an important region for *O. corniculata*. Although this evidence is weak these results are consistent with a native origin of *O. corniculata* in Asia and there is little support for an origin elsewhere.

The spread of *O. corniculata* into the Pacific islands with Polynesian voyagers is also consistent with an east Asian origin. Mitochondrial DNA and Y chromosome studies of Polynesians traces their origins back to Taiwan, the Philippines and Melanesia (*Kayser et al., 2006*; *Ohashi et al., 2006*). Indeed, a proposed spread of *O. corniculata* out of east Asia into the Pacific islands with early colonists could parallel the intentional introduction of *Colocasia esculenta* (L.) Schott (taro) to these island (*Helmkampf et al., 2017*). This, however, does not exclude natural dispersal between islands (*Aoyama, Kawakami & Chiba, 2012*).

*Eiten (1963)* suggested Australasia and south–east Asia as the area of origin of *O. corniculata*. However, he did so for the wrong reasons. He supposed that the variability and broad geographic spread of *O. corniculata* indicated that it was parental to the whole group, rather than a more newly derived species as *Vaio et al. (2013)* have subsequently shown it to be. He also lumped all Australasian specimens into *O. corniculata* s.l.

*Oxalis corniculata* could have been an early introduction to Europe from the east. It may have occurred in Europe before the modern discovery of the New World. An obvious pathway would have been trade along the silk route, which has been suggested as a route of introduction for crops, weeds and plant diseases in both directions (*Wei et al., 2008*; *Smith et al., 2014*). We accept the presence of *O. corniculata* in Europe in the 15th century, but it might have arrived earlier, during the period of the Pax Mongolica (13th–14th centuries) and the reopening of the silk routes at this time. However, the possible earlier presence of *O. corniculata* in Carthage indicates an earlier introduction to the Mediterranean Basin. It is interesting that peach stones also occur in the Carthage deposits. Peaches (*Prunus persica* (L.) Batsch) were introduced to Europe by the 1st century AD and peaches also have an Asian origin (*Faust & Timon, 1995*). Coincidentally, it has also recently been suggested that taro was introduced to Europe in the 4th century AD from south–eastern Asia (*Grimaldi et al., 2018*). Therefore, there are other examples of plants that have been transported between Asia and Europe for agriculture even before the 2nd millennium.

The situation in sub-Saharan Africa is still unclear and may only be resolved once the taxonomic position of *O. radicosa* and *O. procumbens* has been established. In the case of South Africa, Cape Town was originally founded 1652 and was an important trading port for the Dutch East India Company during the 17th and 18th century. As such,

it is not hard to envisage that *O. corniculata* was introduced there. Nevertheless, the testimony of Floras indicates that even if *O. corniculata* was present in Africa before the modern era, it has become more common and today has a wider distribution than previously.

In South America the final difficulty is the status of *O. calachaccensis*, which is indistinguishable morphologically and genetically from *O. corniculata*. Its presence in Bolivia suggests that *O. corniculata* is either native to South America, or that the specimens described as *O. calachaccensis* are in fact introduced *O. corniculata*. Certainly, Knuth described *O. calachaccensis* from plants collected comparatively recently (*Knuth, 1915*). We are inclined to think that specimens described as *O. calachaccensis* are modern introductions and that *O. calachaccensis* should be considered a synonym of *O. corniculata*.

Relying upon physical evidence to determine the native origins of plant has its limitations and biases. A complementary method would be to examine the genetic diversity of *O. corniculata* globally. Taxa tend to be most genetically diverse in the area of their origin (*Sakai et al., 2001*) and such an approach has been useful in understanding the invasion biology of *O. pes-caprae* L. (*Ferrero et al., 2015*). It is hoped that such a study might be possible in the future if the difficulties of collecting representative populations from six continents can be surmounted. Such a study would further define the evolution and migration of this taxon worldwide.

## CONCLUSIONS

While many authors assume that the origin of *O. corniculata* is lost to time, we have shown that there is sufficient evidence to conclude that it is in fact native to East Asia and unlikely to be native to Europe. Many of the details of the global colonization by species of section *Corniculatae* remain a mystery, particularly in Africa and Australasia. Furthermore, it is hoped that this paper will raise awareness of the issues, so that new archaeological discoveries of *O. corniculata* remains are recognised for their importance in understanding the biogeography of this species.

## ACKNOWLEDGEMENTS

The authors would like to thank all the herbaria who helped with this project, including The Royal Botanic Garden, Edinburgh; The Royal Botanic Gardens, Kew and the Museum of Evolution, Uppsala. A special word of thanks also goes to the Biodiversity Heritage Library and the Global Biodiversity Information Facility and their contributors for making such research possible. We would also like to thank the reviewers for their insights and the care they took in reviewing this paper.

### Funding

The authors received no funding for this work.

## Competing Interests

The authors declare that they have no competing interests.

## Author Contributions

- Quentin J. Groom conceived and designed the experiments, performed the experiments, analyzed the data, prepared figures and/or tables, authored or reviewed drafts of the paper, and approved the final draft.
- Jan Van der Straeten performed the experiments, analyzed the data, prepared figures and/or tables, and approved the final draft.
- Ivan Hoste analyzed the data, authored or reviewed drafts of the paper, and approved the final draft.

## Data Availability

Groom, Q. (2018). Earliest records of *Oxalis corniculata* by country (Version 1) [Data set]. Zenodo. DOI 10.5281/zenodo.1332025.

Van der Straeten, J. & Groom, Q. (2018). A bibliography of the medicinal uses of *Oxalis corniculata* [Data set]. Zenodo. DOI 10.5281/zenodo.1254000.

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
