# Peer review of "The origin of Oxalis corniculata L"

_PeerJ, doi:10.7717/peerj.6384_

## Round 0.1 · original submission · Minor Revisions

Dear authors

Your ms has been well received with the overall opinion being for a minor revision.

Regards,

Michael Wink
Academic Editor

Reviewer 1 ·

Basic reporting

The ms. by Groom et al. describes the occurrences of Oxalis corniculata in space and time.
I regard this as a very interesting contribution to the question on the origin of this taxon. I agree with the authors that the question cannot be finally solved (as theoretically all scientific questions). Overall, a refreshing study, relevant for the evaluation of neophytes, in this case worldwide.
The evidence they provide is relatively strong. I like the combination of different approaches and the old illustrations. Still, the phylogenetic data by Vaio where the American natives O. dillenii and stricta also group together with O. corniculata may be a counter-argument. By just looking at the tree an American origin seems more likely. Of course, long-distance dispersal may also have happened before the mrca of Oxalis corniculata from America to Asia. Perhaps the authors should discuss the available phylogenetic data in more detail. Besides the infraspecific problems of O.c., the inclusion of all putative related species in the phylogenetic study by Vaio should be discussed.

A different approach is worth to be mentioned: that a population genetic approach may reveal the origin based on the assumption that the highest genetic diversity represents source areas. Especially for neophytes this makes sense. It has been nicely shown for Oxalis pes-caprae in the Mediterranean with a South African origin.

Discuss the use of seed morphology to distinguish O. species (corniculata from relatives). Best illustrate it with photos. This would strengthen the arguments.

I found some more (secondary) literature:
- Bauhin cites O.c. for Heidelberg around 1600 (publication?).
- Graebner 1914 for Basel (Synop. Mitteleur. Flora)
- Gmelin 1806 (flora badensis) also mentioned it.

After these minor revisions the ms. should be fine for PeerJ.


Minor points:
l. 94, 123, 266, 304 O. => write out Oxalis (at begin of sent.)
l. 95 use => used
l. 137 Earlier? Than…
l. 139 one taxon
l. 145 flowers of this illustration are in bud and are of little diagnostic use. Another herbal BOOK? from circa

Experimental design

see Basic reporting

Validity of the findings

see Basic reporting

Additional comments

see Basic reporting

·

Basic reporting

No comment, except for a suggestion:

There are, in addition to taro, several plant species that currently have wide (cosmopolitan or pantropical) distributions and it may be interesting to refer to some of them (and especially to literature on their historical biogeography, if anything is available). These include Lagenaria siceraria (African origin, but earliest archaeological evidence in Mexico?; China?); Zea mais, Capsicum frutescens, Capsicum chinensis, Sesamum indicum, Pteridium aquilinum, Momordica charantia, Dysphania ambrosioides and Carthamus tinctorius. It appears to be a general problem to determine the origins of plants that are associated with agriculture and human migrations? How did others try to solve the problem?

Experimental design

No comment, except for a suggestion:

Genetic diversity is barely mentioned, yet this may provide an important additional avenue of research. If, for example, all European, African and American collections of Oxalis corniculatus are genetically similar or identical, while the Asian ones are genetically diverse, then the main findings of this paper can be further verified and supported.

Validity of the findings

No comment, except for the suggestion of the potential value of genetic studies as a future direction.

Additional comments

The paper is well conceived and interesting to read, with no obvious errors of fact and/or logic.

There are some very minor errors that can be corrected, perhaps during copy-editing (lines 93, 137, 167, 281, 302, 379 and 441).

---

## Round 0.2 · accepted · Accept

Dear authors

Many thanks for your revision. Good news, your ms can now be accepted.

Kind regards
Michael Wink
Academic Editor

#